# A Recombinant Protein XBB.1.5 RBD/Alum/CpG Vaccine Elicits High Neutralizing Antibody Titers against Omicron Subvariants of SARS-CoV-2

**DOI:** 10.3390/vaccines11101557

**Published:** 2023-10-01

**Authors:** Syamala Rani Thimmiraju, Rakesh Adhikari, Maria Jose Villar, Jungsoon Lee, Zhuyun Liu, Rakhi Kundu, Yi-Lin Chen, Suman Sharma, Karm Ghei, Brian Keegan, Leroy Versteeg, Portia M. Gillespie, Allan Ciciriello, Nelufa Y. Islam, Cristina Poveda, Nestor Uzcategui, Wen-Hsiang Chen, Jason T. Kimata, Bin Zhan, Ulrich Strych, Maria Elena Bottazzi, Peter J. Hotez, Jeroen Pollet

**Affiliations:** 1Texas Children’s Hospital Center for Vaccine Development, Houston, TX 77030, USAmariajose.villarmondragon@bcm.edu (M.J.V.); jslee@bcm.edu (J.L.); zhuyun.liu@bcm.edu (Z.L.); rakhi.tyagikundu@bcm.edu (R.K.); yi-lin.chen@bcm.edu (Y.-L.C.); allan.ciciriello@bcm.edu (A.C.); cristina.poveda@bcm.edu (C.P.); nestor.uzcateguiaraujo@bcm.edu (N.U.); wen-hsiang.chen@bcm.edu (W.-H.C.); bzhan@bcm.edu (B.Z.); bottazzi@bcm.edu (M.E.B.);; 2Department of Pediatrics, National School of Tropical Medicine, Baylor College of Medicine, Houston, TX 77030, USA; 3Department of Molecular Virology and Microbiology, Baylor College of Medicine, Houston, TX 77030, USA; suman.sharma@bcm.edu (S.S.); karm.ghei@bcm.edu (K.G.);; 4Department of Biology, Baylor University, Waco, TX 76706, USA; 5James A. Baker III Institute for Public Policy, Rice University, Houston, TX 77005, USA

**Keywords:** immune escape, vaccine efficacy, COVID-19, SARS-CoV-2

## Abstract

(1) Background: We previously reported the development of a recombinant protein SARS-CoV-2 vaccine, consisting of the receptor-binding domain (RBD) of the SARS-CoV-2 spike protein, adjuvanted with aluminum hydroxide (alum) and CpG oligonucleotides. In mice and non-human primates, our wild-type (WT) RBD vaccine induced high neutralizing antibody titers against the WT isolate of the virus, and, with partners in India and Indonesia, it was later developed into two closely resembling human vaccines, Corbevax and Indovac. Here, we describe the development and characterization of a next-generation vaccine adapted to the recently emerging XBB variants of SARS-CoV-2. (2) Methods: We conducted preclinical studies in mice using a novel yeast-produced SARS-CoV-2 XBB.1.5 RBD subunit vaccine candidate formulated with alum and CpG. We examined the neutralization profile of sera obtained from mice vaccinated twice intramuscularly at a 21-day interval with the XBB.1.5-based RBD vaccine, against WT, Beta, Delta, BA.4, BQ.1.1, BA.2.75.2, XBB.1.16, XBB.1.5, and EG.5.1 SARS-CoV-2 pseudoviruses. (3) Results: The XBB.1.5 RBD/CpG/alum vaccine elicited a robust antibody response in mice. Furthermore, the serum from vaccinated mice demonstrated potent neutralization against the XBB.1.5 pseudovirus as well as several other Omicron pseudoviruses. However, regardless of the high antibody cross-reactivity with ELISA, the anti-XBB.1.5 RBD antigen serum showed low neutralizing titers against the WT and Delta virus variants. (4) Conclusions: Whereas we observed modest cross-neutralization against Omicron subvariants with the sera from mice vaccinated with the WT RBD/CpG/Alum vaccine or with the BA.4/5-based vaccine, the sera raised against the XBB.1.5 RBD showed robust cross-neutralization. These findings underscore the imminent opportunity for an updated vaccine formulation utilizing the XBB.1.5 RBD antigen.

## 1. Introduction

Since the beginning of the COVID-19 pandemic, the SARS-CoV-2 virus has evolved very rapidly, giving rise to a series of mutations, generating multiple variants of concern (VOCs) like Beta, Delta, and, subsequently, Omicron [1]. Within the Omicron lineage, recombinant strains have emerged constantly starting from the earliest BA.1 strain to the latest XBB.1.5, XBB 1.16, and EG.5.1 strains. As of August 2023, these Omicron subvariants appear to have become dominant worldwide. XBB lineages have likely evolved from a recombination event among two BA.2 strains with a mutation at S486P [2]. The additional F486P substitution in XBB.1.5 is believed to offer higher affinity to the human angiotensin-converting enzyme 2 (ACE-2) receptor than seen with BQ.1, XBB, and XBB.1 [3]. As a consequence, immunization or prior exposure to the ancestral wild-type (WT) variant provides suboptimal protection through neutralizing antibodies [4]. Specifically, studies have shown that XBB and other Omicron strains such as BQ.1.1 have higher resistance to humoral immunity induced by vaccination or natural infection than earlier strains like BA.2 and BA.5 [5,6,7]. These findings were confirmed with our recombinant protein vaccines, either RBD219-N1 or RBD203-N1, which encode the receptor-binding domain (RBD) of the SARS-CoV-2 spike protein. In mice and non-human primates, such RBDs, adjuvanted with CpG oligonucleotides and aluminum hydroxide (alum), induced high neutralizing antibodies (Abs) against SARS-CoV-2 (WT) [8,9,10]. Such antigens became central components of the Corbevax vaccine produced by Biological E in India and IndoVac produced by BioFarma in Indonesia, and they have been administered close to 100 million times [11]. 

While these vaccines still offer protection from severe disease, the cross-neutralizing titers against the Omicron strains are suboptimal. As XBB.1.5 is highly resistant to antiviral immunotherapy, the most efficient way to control the current wave is to update vaccine antigens to induce a more effective immunity [12]. Here, we show the development and testing of an XBB.1.5 RBD-based vaccine together with vaccines matching the Beta, Delta, and BA.4/5 variants. We used pseudovirus neutralization assays to determine the cross-protection elicited by these antigens against the ancestral strain (WT) and eight additional variants (Beta, Delta, BA.4, BQ.1.1, BA.2.75.2, XBB.1.16, XBB.1.5, and EG.5.1). 

## 2. Materials and Methods

### 2.1. Sources of Recombinant RBD203-N1 Proteins

The SARS-CoV-2 RBD203-N1 protein was designed and produced as previously described [8,9,10], and it encompasses amino acid residues 332–533 of the SARS-CoV-2 spike protein. The sequence alignment among the RBD variants included in this study is shown in Figure 1. 

### 2.2. Production of the Five Variant RBD Antigens in Pichia Pastoris X-33

To generate the recombinant proteins in yeast, DNAs encoding the SARS-CoV-RBD203-N1 proteins for the WT, Beta, Delta, Omicron BA.4/5 (BA.4 RBD is identical to BA.5), and XBB.1.5 variants were codon-optimized based on yeast codon preference and cloned into the yeast expression vector pPICZαA. The recombinant plasmid DNAs were transformed into *P. pastoris* X33 following a process described previously [8,9,10]. The highest expressing clones for each RBD were used to make research seed stocks that were saved at −80 °C. Fermentation was carried out at the 5 or 1 L scale, respectively, as previously described [13]. Briefly, the seed stocks for each construct were used to inoculate a 0.5 L buffered minimal glycerol (BMG) medium and were grown at 30 °C and 250 rpm until an OD600 of 5–10. This culture was used to inoculate a sterile low-salt medium (LSM, pH 5.0) with PTM1 trace elements and d-Biotin. Cell expansion was continued at 30 °C with a dissolved oxygen (DO) set point of 30% until glycerol depletion. Then, methanol was pumped in from 1 mL/L/h to 11 mL/L/h over a 6 h period; the pH was adjusted to 6.5. The methanol induction was maintained at 11 mL/L/h at 25 °C for 70 h, except for XBB.1.5-RBD, which was induced for 48 h. After fermentation, the culture was harvested by centrifugation, filtered, and kept frozen at −80 °C until purification. The recombinant RBD protein was captured from the fermentation supernatant using a butyl Sepharose high-performance column (Cytiva) in the presence of ammonium sulfate salt at a concentration of 0.8 M (XBB.1.5-RBD) or 1.1 M (WT-RBD, Beta-RBD, Delta-RBD, and BA.4/5-RBD) in HIC buffer (30 mM of Tris-HCl and pH 8.0). RBD protein was further purified using a Q Sepharose XL (QXL) column (Cytiva) in a negative mode in QXL buffer A (20 mM of Tris-HCl and pH 7.5) with 100 mM of NaCl (WT-RBD and Beta-RBD), 50 mM of NaCl (Delta-RBD), or 0 mM of NaCl (XBB.1.5-RBD). For BA.4/5-RBD, a negative QXL chromatography was performed in QXL buffer B (20 mM of Tris-HCl, pH 8.4, and 10 mM of NaCl) followed by dialysis in QXL buffer C (20 mM of L-histidine, 100 mM of NaCl, and pH 6.0) for storage. All RBD proteins were aseptically filtered using a 0.22 µm filter and stored at −80 °C until usage. To evaluate these RBD proteins’ biophysical characteristics and functionality, SDS-PAGE-based densitometry, dynamic light scattering, and ACE-2 binding assays were performed following the methods previously described [14]. Size Exclusion High Pressure Chromatography (SE-HPLC) was performed by injecting 50 µg of RBD on an XBridge Premier Protein SEC Column (Waters, Cat# 186009959) connected with a corresponding guard column (Waters, Cat# 186009969). The protein was eluted with 1X TBS, pH 7.5, at a 0.5 mL/min flow rate. The Bio-Rad gel filtration standard (Bio-Rad: Hercules, CA, USA, catalog# 1511901) was used as a control.

### 2.3. Vaccine Formulations and Preclinical Study Design

Animals were housed and were provided care in strict adherence to the guidelines set forth by local, state, federal, and institutional policies. Facilities were accredited by AAALAC International, meeting the standards outlined in the Animal Welfare Act and the Guide for the Care and Use of Laboratory Animals. Experiments were performed under an approved protocol from the Baylor College of Medicine Institutional Animal Care and Use Committee. Female BALB/c mice (N = 8/group), aged 6–8 weeks old, were immunized twice intramuscularly at 21-day intervals with the antigens shown in Appendix A and then euthanized two weeks after the second vaccination. Each dose of the SARS-CoV-2 RBD vaccine contained 7 µg of RBD protein, 200 µg of alum (*Alhydrogel*^®^, aluminum hydroxide, Catalog # AJV3012, Croda Inc., Snaith, UK), and 20 µg of CpG1826 (Invivogen, San Diego, CA, USA). The SARS-CoV-2 RBD proteins were prepared with 1xTBS buffer (20 mM of Tris, 100 mM NaCl, and pH 7.5). Before injection, alum and CpG 1826 were added, and the sample was vortexed for 3 s. 

### 2.4. Serological Antibody Measurements Using ELISA

To examine SARS-CoV-2 RBD-specific antibodies in the mouse sera, indirect ELISAs were conducted as published before [9]. Briefly, plates were coated with 0.2 µg/well of SARS-CoV-2 RBD proteins from different variants. Mouse sera samples were 3-fold diluted from 1:200 to 1: 11,809,800 in 0.1% BSA in an assay buffer (0.05% Tween20 in 1x PBS). Samples were prepared and measured in duplicate. Assay controls included a 1:400 dilution of pooled naïve mouse sera (negative control), 1:10,000 dilution of pooled high titer mouse (positive control), and assay buffer as blanks. A total of 100 µL/well of 1:6000 goat anti-mouse IgG HRP in assay buffer was added. After incubation, plates were washed five times, followed by adding 100 µL/well of TMB substrate. Plates were incubated for 15 min at room temperature (RT) while protected from light. After incubation, the reaction was stopped by adding 100 µL/well of 1 M of HCl. The absorbance at a wavelength of 450 nm was measured using a BioTek Epoch 2 spectrophotometer. For each sample, the titer was determined using a four-parameter logistic regression curve of the absorbance values. The titer cutoff value: negative serum control + 3 x standard deviation of the negative serum control. 

### 2.5. Pseudovirus Assay for Determination of Neutralizing Antibodies

To test for neutralizing antibodies, we prepared non-replicating lentiviral particles expressing the SARS-CoV-2 spike variant proteins on their envelope membrane and encoding luciferase as a reporter. Infection was quantified using in vitro-grown human 293 T-hACE-2 cells based on luciferase expression. The pseudovirus production system included the luciferase-encoding reporter plasmid, pNL4-3, lucR-E-, a Gag/Pol-encoding packaging construct (pΔ8.9), and the codon-optimized SARS-CoV-2 spike VOC-expressing plasmids (pcDNA3.1-CoV-2 S gene), based on clone p278-1 [9]. Pseudovirus-containing supernatants were recovered 48 h after transfection, passed through a 0.45 µm filter, and saved at −80 °C until used for neutralization studies. 

The expression plasmids for SARS-CoV-2 spike variants of concern WT, Beta, and Delta were generated by site-directed mutagenesis or replacement of segments of the codon-optimized Wuhan SARS-CoV-2 spike expression clone, p278-1, with variant sequences as previously described [8]. Omicron spike variants BA.4, BA.2.75.2, BQ.1.1, XBB.1.5, and EG.5 sequences were generated by changing codons of the p278-1 spike clone sequence to produce the consensus amino acid sequence of each variant. The variant spike sequences were synthesized with the 3′ Flag-tag (Genscript: Piscataway Township, NJ, USA) and inserted into the pcDNA3.1 expression vector. The list with the variant-specific mutations added to the spike protein variant clones can be found in the Appendix A. The sequences of all the variant spike genes were confirmed via commercial DNA sequencing.

The pseudovirus assay was performed as described earlier [9]. Briefly, 10 μL of pseudovirus was incubated with serial dilutions of the serum samples for 1 h at 37 °C. Next, 100 µL of sera-pseudovirus were added to 293 T-hACE-2 cells in 96-well poly-D-lysine-coated culture plates. Following 48 h of incubation in a 5% CO_2_ environment at 37 °C, the cells were lysed with 100 µL of Promega Glo Lysis buffer for 15 min at room temperature. Finally, 50 µL of the lysate was added to 50 µL luciferase substrate (Promega Luciferase Assay System). The amount of luciferase was quantified with luminescence (relative luminescence units (RLU)), using the Luminometer (*Biosynergy* H4, BioTek). Pooled sera from vaccinated mice (*n* = 8) were compared by their 50% inhibitory dilution (IC50), defined as the serum dilution at which the virus infection was reduced by 50% compared to the negative control (virus + cells). IC50 values were calculated as described by Nie et al. [15]. Samples were measured in duplicate. Statistical analyses were performed on sets of IC50 titers elicited using sera from mice vaccinated with each of the RBD vaccines against pseudovirus variants using GraphPad Prism 8.0 to rank the RBD vaccines according to Spearman’s correlation coefficient. 

## 3. Results

### 3.1. Expression, Purification, and Characterization of Recombinant Proteins of Different Variants of SARS-CoV-2 RBD203

After *P. pastoris* X-33 plasmid transformation, recombinant proteins of different variants of the SARS-CoV-2 RBD, including WT, Beta, Delta, Omicron BA.4/5, and Omicron XBB.1.5, were expressed by induction with methanol at 30 °C for 72 h (WT, Beta, Delta, and BA.4/5) or 48 h (XBB.1.5) in either a 1 or a 5 L fermentation process. Following the published protocol for RBD203-N1 [13], all proteins were purified using a combination of a hydrophobic interaction and ion exchange chromatography. While all proteins were purified to more than 96% homogeneity, their yields differed (Table 1). In particular, the RBDs of the two Omicron strains, BA.4/5 and XBB.1.5, showed reduced yields. We are currently optimizing the purification conditions of these strains to prepare for technology transfer to manufacturing partners. 

When analyzed using SE-HPLC, all RBD proteins produced a single peak detected at UV280 nm, confirming the homogeneity of each preparation (Figure 2A). Moreover, the apparent molecular weight via SE-HPLC matched that observed with densitometry scanning of SDS-PAGE gels (Table 1).

The functionality of the purified RBD molecules was established in an ACE-2 binding assay (Figure 2B), where all proteins were shown to bind recombinant human ACE-2. Consistent with prior data published by Mannar et al. [16], and the affinity of the Omicron RBDs seemed to be higher than the other RBD variants, while the WT-RBD possessed the lowest binding affinity to ACE-2. 

### 3.2. Immune Response Induced by Adjuvanted RBD Vaccines in Mice

Preclinical studies were performed in eight to nine-week-old female BALB/c mice, using yeast-produced SARS-CoV-2 recombinant RBD proteins, adjuvanted with aluminum hydroxide and CpG1826. Mice (*n* = 8/group) were immunized twice intramuscularly on days 0 and 21 with a dose of 7 µg of RBD, 200 µg of alum, and 20 µg of CpG. On day 35, the study was terminated, and the sera were collected and evaluated for total RBD-specific IgG and neutralizing antibodies against pseudoviruses matching the WT SARS-CoV-2 isolate, Beta, Delta, as well as the Omicron variants BA.4 and XBB.1.5 (Figure 3).

We found that the total RBD-specific IgG antibody titers elicited in mouse sera with all the five (WT, Beta, Delta, BA.4/5, and XBB.1.5) RBDs were high when tested with immobilized antigens in an ELISA. (Figure 3A). However, when sera were tested for neutralizing antibodies in a pseudovirus assay, we observed that cross-protection of WT-RBD-induced sera against the Omicron BA.4/5 pseudovirus was minimal (Figure 3B), with a 73-fold reduction in the IC50 from 4174 against WT pseudovirus to an IC50 of 57 against the BA.4/5 pseudovirus. The titer against the Omicron XBB.1.5 pseudovirus was even lower, with an IC50 of 29. Nab titers induced using Beta RBD and Delta RBD also failed to suggest protection against Omicron XBB.1.5 pseudovirus (Appendix A). However, sera raised against the BA.4/5 RBD showed no protection against WT (IC50 = 20), Beta (IC50 = 193), and Delta (IC50 = 20) pseudoviruses, and notably, the sera offered little protection against XBB.1.5 (IC50 = 40) pseudoviruses. Mice vaccinated with the XBB.1.5 RBD vaccine showed low neutralizing antibody titers against the WT, Beta, and Delta pseudoviruses as reported earlier [17]. By contrast, they demonstrated high titers of neutralizing antibodies against the Omicron BA.4 and XBB.1.5 pseudoviruses (Figure 3B). Interestingly, protection against BA.4/5 pseudovirus was even more pronounced with the XBB.1.5 vaccine than with the homologous antigen (Figure 3B), an observation that is similar to what was seen in the high anti-BA.4/5 neutralizing antibody titers generated by the Moderna XBB.1.5 vaccine [18].

In a further expansion of our cross-neutralization studies, we tested the same sera against the Omicron pseudoviruses: BA.2.75.2, BQ.1.1, BA.4, XBB.1.5, XBB.1.16, as well as EG.5.1 (Figure 4). We observed a significant deficiency in cross-neutralization for most of the mouse sera from animal vaccinated with either WT-RBD, Beta-RBD, or Delta-RBD when confronted with the Omicron variants. Notably, even for sera from mice vaccinated with the BA.4/5 vaccine, evidence of immune evasion was observed. These results are in accordance with data reported by other researchers [17]. In contrast, the sera from mice vaccinated with the updated vaccine comprising the XBB.1.5-RBD antigen exhibited exceptional cross-neutralization capabilities against all tested Omicron spike pseudoviruses. Particularly encouraging was the remarkably high IC50 value (IC50 = 5758) exhibited by the XBB1.5 sera against the EG.5.1 pseudovirus. This is especially significant as EG.5.1 is among the current emerging variants of concern, rapidly spreading worldwide.

Using a Spearman correlation on the IC50 values, we illustrate the number of different neutralizing epitopes comparing the RBD antigens of SARS-CoV variants. While we noticed a positive gradient of similarity (Figure 5, Blue) among the RBDs of three earlier variants of concern (WT, Beta, and Delta), we found a negative correlation coefficient value comparing WT, Beta, and Delta RBDs to Omicron BA.4/5 and XBB.1.5 RBDs (Figure 5, Red). Likewise, Omicron RBD BA.4/5 and XBB.1.5 share many neutralizing epitopes, while no correlation was found with the earlier RBDs.

## 4. Discussion

In the fourth year of the pandemic, immune escape variants of SARS-CoV-2 continue to pose a significant threat to global health. Omicron variants such as BA.2.75.2, BQ.1.1, XBB.1.5, XBB.1.16, and EG.5.1 are characterized by a large number of mutations with increased transmissibility and more pronounced immune evasion. As these new variants emerge, intra-VOC recombination events are commonly observed, leading to an ever-increasing cascade of sub-lineages. Current data indicate that the XBB.1.5 strain is highly resistant to monoclonal antibodies and convalescent plasma treatment [8,9,10]. The XBB subvariant’s high transmissibility and its high number of mutations likely have contributed to immune evasion. Thus, the vaccines that only targeted the ancestral virus are not as successful in raising neutralizing antibodies against more recent subvariants. In addition, bivalent boosters that contain both ancestral and BA.4/5 antigens are poorly neutralizing against Omicron-derived next-generation subvariants, including XBB.1.5 [19]. Therefore, the broader neutralizing immune response generated by the XBB.1.5 antigen against additional Omicron subvariants in this study is encouraging for protection against current circulating strains and potential future variants.

In this study, we compared the efficacy of various vaccines based on the RBD of variant SARS-CoV-2 spike proteins, with the goal of better characterizing cross-neutralization. We observed minimal heterologous cross-neutralization of the XBB.1.5 pseudovirus with immune sera generated with vaccines based on RBDs of early VOCs. Even immunizations with the more recent BA.4/5 RBD generated only partial cross-neutralization of the XBB.1.5 pseudovirus. These results mirror findings for breakthrough human infections of BA.4, which did not generate adequate cross-neutralizing antibodies to protect against XBB subvariants [20]. Our results also re-emphasized that, as compared to the cross-neutralization of the earlier three variants (i.e., WT, Beta, and Delta), Omicron variants have evolved enough to successfully escape neutralization using parental vaccines. This was exemplified by the neutralization data from WT, Beta, and Delta RBD vaccinations against both BA.4 and XBB.1.5 pseudoviruses in this study. Importantly, the IgG antibody binding data from ELISAs demonstrated that vaccination with each of the ancestral RBDs (i.e., WT, Beta, and Delta) induced high levels of IgG that could recognize all five of the different RBDs, including BA.4/5 and XBB.1.5. Yet, the sera had low functional neutralizing antibody titers against the BA.4/5 and XBB variant pseudoviruses. We acknowledge that no live virus neutralization assays have been conducted yet with the XBB.1.5 vaccine. However, in our previous publication introducing an earlier version of the RBD vaccine [8], we have shown that, generally, our pseudovirus assay is very well correlated with the live virus PRNT assay (R^2^ = 0.9274). We also note that the WHO and others have shown excellent comparability between pseudovirus and live virus assays [21,22]. 

In addition to the reported immune imprinting caused by previous vaccinations [18], the antigenic diversities caused by mutations on spike proteins [23] between a variety of Omicron sub-variants may further dampen the neutralization process and reduce the immune response against future variants. This is especially true in the case of XBB sub-lineages, where the neutralizing antibodies elicited against XBB sub-variants were very low after BA.5 infection, indicating the evolution of the XBB sub-lineage further away from BA.4/5 [20]. We have observed weak cross-neutralization from BA.4/5 RBD-generated sera against XBB.1.5 and XBB.1.16 subvariants, while the inverse combination of an XBB.1.5 RBD antigen against BA.4/5 pseudovirus offered very strong neutralizing antibody titers. The current study shows the value of the XBB.1.5 RBD antigen with its increased number of mutations in its RBD as a better vaccine candidate with successful cross-neutralization capacity against all subvariants of Omicron such as BA.4/5, BA.2.75, BQ.1.1, XBB.1.16, and XBB.1.5 as well as against the most recent COVID-19 variants such as EG.5.1. 

As the 2023 fall respiratory season approaches, we are seeing an uptick in illnesses caused by the Omicron variant EG.5 or ‘Eris’, and cases could continue to rise [24]. The spike protein of EG.5 shares an almost identical amino acid profile with XBB.1.5, distinguishing itself with an additional F456L amino acid mutation. Within the EG.5 lineage, EG.5.1 is the dominant subvariant (more than 88% of cases) [25], which is characterized by an additional spike mutation: Q52H. At the end of August 2023, EG.5 had been linked to 20.6% of COVID-19 cases in the United States, surpassing all other circulating SARS-CoV-2 strains in prevalence [25]. Our data strongly indicate that the RBD XBB1.5 vaccine provides effective protection against the EG.5.1 strain of the SARS-CoV-2 virus through the production of neutralizing antibodies. Such levels of neutralization against those same variants were not imparted by the BA.4/5 RBD vaccine, underscoring the importance of developing the next generation of COVID-19 vaccines customized toward XBB subvariant sequences. Further experiments will focus on using an XBB.1.5 boost after vaccination with the WT and BA.4/5 RBDs to mimic the real-world scenario and yield valuable information about the cross-protection levels against current and potential future versions of XBB derivatives. In addition, we believe that our observation of the broad protection levels of the XBB.1.5 vaccine is also likely applicable to the updated mRNA vaccines from Pfizer and Moderna that also express the XBB.1.5 antigen [12] and are becoming available to the general public in the United States of America by the end of September 2023. 

## Figures and Tables

**Figure 1 vaccines-11-01557-f001:**
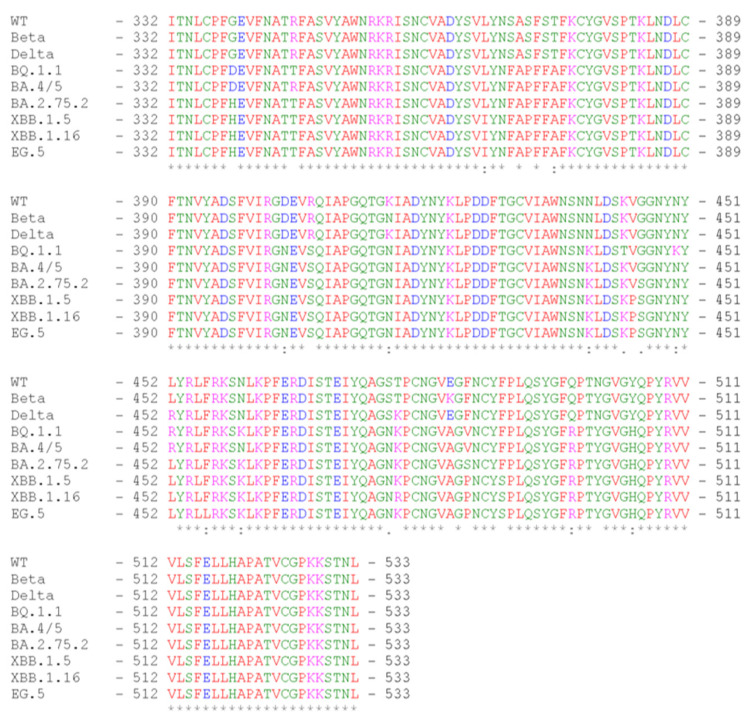
Amino acid sequence alignment of the nine RBD variants investigated in this study (alignment made with https://www.ebi.ac.uk/Tools/msa/clustalo/ (accessed on 28 September 2023)).

**Figure 2 vaccines-11-01557-f002:**
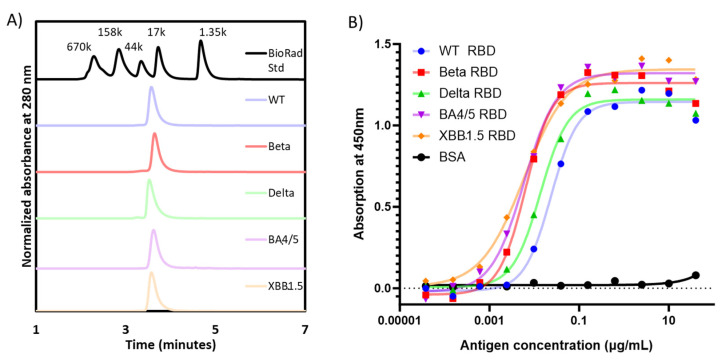
(**A**) SE-HPLC and (**B**) ACE-2 binding ELISA were performed to evaluate the identity/homogeneity and the functionality of five recombinant RBD proteins. The dashed line indicates the lower limit of detection.

**Figure 3 vaccines-11-01557-f003:**
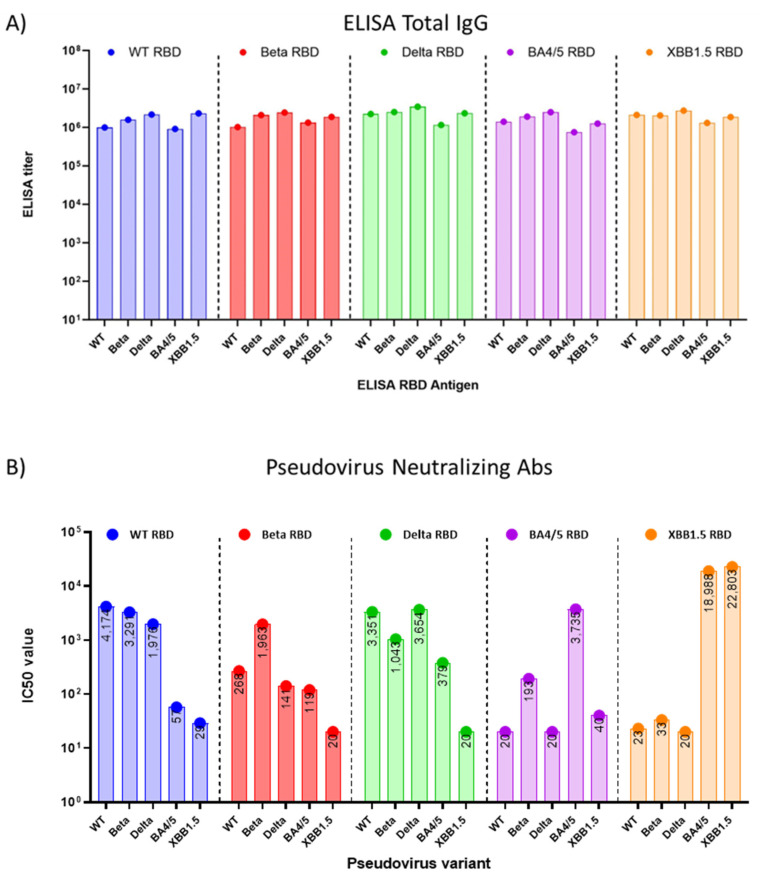
(**A**) Average (*n* = 2) total RBD-specific IgG titers of pooled sera from mice (*n* = 8) immunized with five different RBD vaccine antigens; (**B**) average (*n* = 2) neutralizing antibody titers of pooled sera from mice (*n* = 8) vaccinated with one of five RBD antigens tested against the same five pseudovirus variants. The Y-axis shows IC50 values of the neutralizing antibody titer on a log 10 scale (Appendix A), while the X-axis represents five RBD sera panels, each showing the neutralization titers against five pseudoviruses.

**Figure 4 vaccines-11-01557-f004:**
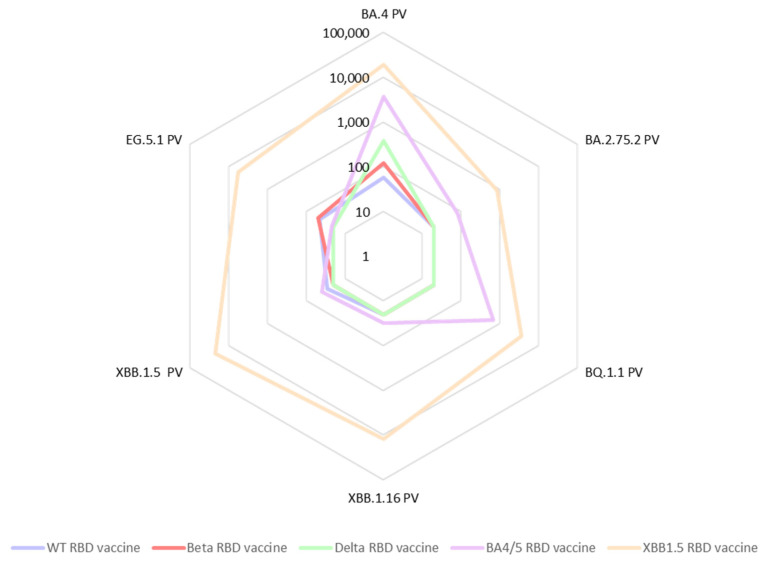
Rader plot of average IC50 values (*n* = 2) of pooled sera from vaccinated mice (*n* = 8), tested against a panel of six Omicron SARS-CoV-2 pseudoviruses (XBB.1.5, BA.4/5, XBB.1.16, BQ.1.1, BA.2.75.2, and EG.5.1). Sera are collected from mice that were immunized twice with one of five recombinant RBD vaccines adjuvanted with alum + CpG.

**Figure 5 vaccines-11-01557-f005:**
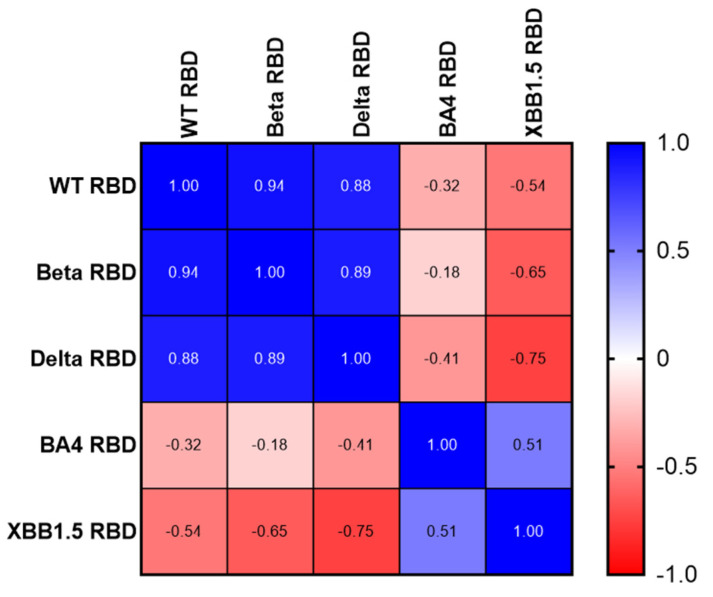
Non-parametric rank matrix of five variant RBD vaccines based on Spearman’s rank correlation coefficients that were calculated based on the IC50 values against different SARS-CoV-2 pseudovirus variants.

**Table 1 vaccines-11-01557-t001:** Biophysical characterization of five recombinant RBD proteins.

RBD Variant	Purification Yield [RBD (mg)/Fermentation Supernatant (L)]	SDS-PAGE with Coomassie Blue Staining (Non-Reduced)		Homogeneity via SE-HPLC (%)	Radius via Dynamic Light Scattering (nm)
		MW (kDa)	Purity (%)		
WT-RBD	270.5	25.3	94.1	99.7	2.57 ± 0.01
Beta-RBD	204.3	23.8	93.2	99.7	2.63 ± 0.01
Delta-RBD	294.5	25.1	93.9	98.4	2.73 ± 0.02
BA.4/5-RBD	46.2	24.7	94.2	96.4	2.58 ± 0.02
XBB.1.5-RBD	78.0	25.1	93.2	99.4	2.71 ± 0.03

## Data Availability

The data that support the findings of this study are available on request from the corresponding author.

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
