# Peer review of "A Recombinant Protein XBB.1.5 RBD/Alum/CpG Vaccine Elicits High Neutralizing Antibody Titers against Omicron Subvariants of SARS-CoV-2"

_vaccines, 2023, doi:10.3390/vaccines11101557_

Round 1

Reviewer 1 Report

In this manuscript, the authors report the development and characterization of a next-generation vaccine adapted to SARS-CoV-2 XBB variants. Although the total RBD-specific IgG titer profiles look similar among five RBD variants, the neutralizing antibody titer profile or cross-protection profile against the pseudoviruses looks very different between XBB1.5 RBD and other additional four RBD variants (WT, Beta, Delta, BA.4/5).The conclusion that low levels of neutralization elicited by current hybrid immunity after BA.5 infection against XBB lineages may suggest an urgent need for the development of next-generation COVID-19 vaccines that can provide a protection for future variants. The results are very interesting and useful for vaccination strategy development. The rationale and objective are clear and addressed in the Introduction. The methods are well stated in the Materials and Methods. And the results are also well discussed in the Discussion. 

The following items are intended to improve the quality of the manuscript:

1.     “wild-type (WT)” may only be used once through the entire manuscript. “WT” or “wild-type” can replace “wild-type (WT)” in other sentences if “wild-type (WT)” initially appeared, in addition to “ancestral strain (WT)”.

2.     The sentence “The XBB.1.5 27 RBD/CpG/alum vaccine induced robust antibody production in mice that demonstrated strong neutralization of the XBB.1.5 pseudovirus and multiple other Omicron pseudoviruses” sounds confused. In science, there may be no direct correlation between “robust antibody production” and “that demonstrated strong neutralization”.

3.     Suggest modifying then adding the following statement into “2.3. Vaccine formulations and preclinical study design”: Animals were housed and cared for in accordance with local, state, federal and institutional policies in facilities accredited by AAALAC International under standards established in the Animal Welfare Act and the Guide for the Care and Use of Laboratory Animals.

4.     Suggest adding statistic analysis method in the Materials and Methods.

5.     “total IgG” in lines 202, 205 and other lines may be better descripted as “total RBD-specific IgG”.

6.     Looks like no variability within the group data in Figures 3A and 3B although some deviations are showed in Figure S1. An explanation of statistical analysis within the group data would be helpful, as the "bars" instead of "individual points" per group is very apparent relative to the other panel data. In addition, the number of serum samples per group (n=?) used in the assays should be stated in the legends of Figures 3A and 3B.

7.     Since this study is for the development of a translational vaccine, the evaluation of the vaccine protective efficiency against authentic SARS-CoV-2 variants such as XBB1.5 variant would also be helpful. The authors may add an explanation message in the Discussion if such efficiency data are not available.

The manuscript is well-written, but there are several occurrences such as some typing errors and grammatical or meaningful confusion that require a careful proof prior to publication.

Author Response

Reviewer 1:

In this manuscript, the authors report the development and characterization of a next-generation vaccine adapted to SARS-CoV-2 XBB variants. Although the total RBD-specific IgG titer profiles look similar among five RBD variants, the neutralizing antibody titer profile or cross-protection profile against the pseudoviruses looks very different between XBB1.5 RBD and other additional four RBD variants (WT, Beta, Delta, BA.4/5).The conclusion that low levels of neutralization elicited by current hybrid immunity after BA.5 infection against XBB lineages may suggest an urgent need for the development of next-generation COVID-19 vaccines that can provide a protection for future variants. The results are very interesting and useful for vaccination strategy development. The rationale and objective are clear and addressed in the Introduction. The methods are well stated in the Materials and Methods. And the results are also well discussed in the Discussion. 

The following items are intended to improve the quality of the manuscript:

  1. “wild-type (WT)” may only be used once through the entire manuscript. “WT” or “wild-type” can replace “wild-type (WT)” in other sentences if “wild-type (WT)” initially appeared, in addition to “ancestral strain (WT)”.

Response: The comment was addressed in the updated manuscript.

  1. 2.   The sentence “The XBB.1.5 27 RBD/CpG/alum vaccine induced robust antibody production in mice that demonstrated strong neutralization of the XBB.1.5 pseudovirus and multiple other Omicron pseudoviruses” sounds confused. In science, there may be no direct correlation between “robust antibody production” and “that demonstrated strong neutralization”.

Response: Thank you for pointing this out. We rewrote the sentence as follows: “The XBB.1.5 RBD/CpG/alum vaccine elicited a robust antibody response in mice. Furthermore, serum from vaccinated mice demonstrated potent neutralization against the XBB.1.5 pseudovirus as well as several other Omicron pseudoviruses.”

  1. Suggest modifying then adding the following statement into “2.3. Vaccine formulations and preclinical study design”: Animals were housed and cared for in accordance with local, state, federal and institutional policies in facilities accredited by AAALAC International under standards established in the Animal Welfare Act and the Guide for the Care and Use of Laboratory Animals.

Response: Thank you for that suggestion. The following statement was added: “Animals were housed and provided care in strict adherence to the guidelines set forth by local, state, federal, and institutional policies. Facilities were accredited by AAALAC International, meeting the standards outlined in the Animal Welfare Act and the Guide for the Care and Use of Laboratory Animals. Experiments were performed under a Baylor College of Medicine Institutional Animal Care and Use Committee-approved protocol.”

  1. Suggest adding statistic analysis method in the Materials and Methods.

We have added additional information in the Materials and Methods section to describe the statistical analysis of the ELISA data and the pseudovirus neutralization data. We also added a line under Figure 3 to indicate the sample size of the data shown in the bar graphs.

  1. “total IgG” in lines 202, 205 and other lines may be better descripted as “total RBD-specific IgG”.

Response: Thank you for that suggestion.  We have revised the manuscript as proposed by the reviewer.

  1. Looks like no variability within the group data in Figures 3A and 3B although some deviations are showed in Figure S1. An explanation of statistical analysis within the group data would be helpful, as the "bars" instead of "individual points" per group is very apparent relative to the other panel data. In addition, the number of serum samples per group (n=?) used in the assays should be stated in the legends of Figures 3A and 3B.

Response: To be more specific on how the data was obtained, we have revised the captions of Figures 3A and 3B as follows: “Average (n=2) total RBD-specific IgG titers of pooled sera from mice (n=8) immunized with five different RBD vaccine antigens, B) Average (n=2) neutralizing antibody titers of pooled sera from mice (n=8) vaccinated with one of five RBD antigens tested against the same five pseudovirus variants.”

  1. Since this study is for the development of a translational vaccine, the evaluation of the vaccine protective efficiency against authentic SARS-CoV-2variants such as XBB1.5 variant would also be helpful. The authors may add an explanation message in the Discussion if such efficiency data are not available.

Response: Thank you for your comment. Virus neutralization assays using the pseudovirus platform offer practical and ethical advantages while still providing valuable insights into vaccine efficacy and immune responses. In our previous publication (Pollet et al., 2022, Vaccine, 40, 25, 3655-3663), we utilized a live virus assay to demonstrate neutralization for a prior version of the same vaccine mentioned in the current manuscript under review. In both instances, we employed the same pseudovirus neutralization assay for analyzing the mouse sera, and we meticulously showcased a strong correlation between the IC50 values of our pseudovirus assay and the live virus PRNT (Fig. 3c, R2=0.9274). We have updated the manuscript to reference this prior publication and highlight the correlation between our pseudovirus and PRNT assays.

Virus challenge studies in animal models demand significant financial resources for the acquisition and maintenance of laboratory animals, specialized equipment, and trained personnel. While we acknowledge the necessity of these studies in evaluating vaccine candidates before clinical testing, their inherent constraints often result in prolonged timelines. It is our strong belief that this first dataset should be expediently shared within the research community to provide timely insights into the advantages of vaccination with updated SARS-CoV-2 antigens.

Reviewer 2 Report

The manuscript titled "A recombinant protein XBB.1.5 RBD/Alum/CpG vaccine elicits 2 high neutralizing antibody titers against Omicron subvariants 3 of SARS-CoV-2" describes the development and characterization of a next-generation vaccine adapted to the recently emerging XBB variants of SARS-CoV-2 by conducted preclinical studies in mice using a novel yeast-produced SARS-CoV-2 XBB.1.5 RBD subunit vaccine candidate formulated with alum and CpG.  Modest cross-neutralization against Omicron subvariants by sera from mice vaccinated with the original RBD/CpG/Alum vaccine or with the BA.4/5-based vaccine was obtained. However, sera raised against the XBB.1.5 RBD showed robust cross-neutralization. These findings underscore the opportunity for an updated vaccine formulation utilizing the XBB.1.5 RBD antigen.

The manuscript is a valuable addition to the clinical data available and has significance in mitigating the issues associated with coronaviruses.

No major comments

Author Response

Thank you very much for taking the time to review this manuscript.  We appreciate your supportive comments. 

Reviewer 3 Report

In the manuscript entitled 'A recombinant protein XBB.1.5 RBD/Alum/CpG vaccine elicits high neutralizing antibody titers against Omicron subvariants of SARS-CoV-2' by Thimmiraju et al., the authors developed recombinant protein vaccines for SARS-CoV-2 using different variant sequences of RBD. They characterized the expression and ACE-2 binding of the recombinant RBDs and studied the binding activity and neutralization titers to variants of the SARS-CoV-2 pseudovirus of immunized mice sera. They showed that the XBB1.5 RBD vaccine was able to elicit higher neutralizing antibody titers against XBB sub-strains. This manuscript aims to answer an important question on the cross-activity of the XBB1.5 RBD vaccine toward earlier variants. However, the results present in this paper were not enough to support the conclusions, and several concerns are raised below:

1.     In this study, the authors aim to investigate the level of neutralizing antibody titer elicited by the XBB1.5 RBD vaccine against SARS-CoV-2 variants. However, only binding ELISA and pseudoviral neutralization assays were used here. Given the development of SARS-CoV-2 vaccine studies, many methods to measure humoral immune responses are widely used. The authors need to include at least some SARS-CoV-2 live virus variants to further investigate the neutralization titers.

2.     There have been many novel designs to improve RBD vaccine immunogenicity, such as dimer or trimer format, etc. The recombinant protein RBD vaccine proposed here is similar to what has been published or even approved in the clinic. This design makes this study less novel.

3.     In Figure 3B, it's surprising to see that the XBB1.5 RBD vaccine elicits a higher neutralizing antibody titer towards BA4 than the BA4/5 RBD vaccine. The authors should include additional assays to answer this question or give an in-depth discussion on this observation using available literature.

4.     In Figure 4, the neutralizing antibody titer of the BA4/5 RBD vaccine is missing. Please include that.

 Minor editing of the English language is required for this manuscript. 

Author Response

Reviewer 3:

In the manuscript entitled 'A recombinant protein XBB.1.5 RBD/Alum/CpG vaccine elicits high neutralizing antibody titers against Omicron subvariants of SARS-CoV-2' by Thimmiraju et al., the authors developed recombinant protein vaccines for SARS-CoV-2 using different variant sequences of RBD. They characterized the expression and ACE-2 binding of the recombinant RBDs and studied the binding activity and neutralization titers to variants of the SARS-CoV-2 pseudovirus of immunized mice sera. They showed that the XBB1.5 RBD vaccine was able to elicit higher neutralizing antibody titers against XBB sub-strains. This manuscript aims to answer an important question on the cross-activity of the XBB1.5 RBD vaccine toward earlier variants. However, the results present in this paper were not enough to support the conclusions, and several concerns are raised below:

  1. In this study, the authors aim to investigate the level of neutralizing antibody titer elicited by the XBB1.5 RBD vaccine against SARS-CoV-2 variants. However, only binding ELISA and pseudoviral neutralization assays were used here. Given the development of SARS-CoV-2 vaccine studies, many methods to measure humoral immune responses are widely used. The authors need to include at least some SARS-CoV-2 live virus variants to further investigate the neutralization titers.

Response: Thank you for your comment - we agree that live virus assays are important and are historically the gold standard when it comes to evaluating functional antiviral immune responses. In fact, in our prior publication (Pollet et al., 2022, Vaccine, 40,25, 3655-3663) we have used PRNTs to demonstrate neutralization for a prior version of the same vaccine introduced in the manuscript under review here. In that previous publication, we also employed the same pseudovirus neutralization assay that we used for the analysis of the XBB.1.5 sera in the new manuscript. We took great care to show that our pseudovirus assay IC50 values are well correlated with our live virus PRNT (Fig.3c, R2=0.9274). We have added a line to the manuscript referencing that prior publication and the correlation of our pseudovirus assay with our PRNT assay. 

We also note that the WHO in its technical guidance for antigenic SARS-CoV-2 monitoring from June 2022 states that pseudovirus assays for SARS-CoV-2 have been shown to correlate well with those from live virus assays (Sholukh et al., 2021, J Clin Microbiol, 20,59, e0052721). Robustness and comparability of SARS-CoV-2 pseudovirus assays were further shown by, for example, Riepler et al., 2020, Vaccines, Basel, 9(1), 13. We, therefore, see no reasonable possibility that a live virus assay for the XBB.1.5 vaccine would yield any different result than the pseudovirus assay. It would, however, add significant costs to the project and delay the publication of what we consider is currently relevant data for the appreciation of the new XBB.1.5 vaccines.

  1. There have been many novel designs to improve RBD vaccine immunogenicity, such as dimer or trimer format, etc. The recombinant protein RBD vaccine proposed here is similar to what has been published or even approved in the clinic. This design makes this study less novel.

Response: We appreciate the reviewer's perspective, and we do not claim to possess the most groundbreaking vaccine platform. However, our platform offers a valuable advantage by enabling the production of safe and cost-effective vaccines at a large scale, as demonstrated in our prior work (Pollet et al., 2021, Adv Drug Deliv Rev. Mar;170:71-82). This capability aligns with our commitment to serving the global community's vaccination needs. (Hotez et al., 2023, "From concept to delivery: a yeast-expressed recombinant protein-based COVID-19 vaccine technology suitable for global access," Expert Review of Vaccines, 22:1, 495-500). Additionally, we are confident that our data pertaining to the XBB1.5 version of the vaccine holds significant merit, providing further evidence of the advantages conferred by vaccination with updated SARS-CoV-2 antigens.

  1. In Figure 3B, it's surprising to see that the XBB1.5 RBD vaccine elicits a higher neutralizing antibody titer towards BA4 than the BA4/5 RBD vaccine. The authors should include additional assays to answer this question or give an in-depth discussion on this observation using available literature.

Response: We thank the reviewer for this observation. While the result appears counter-intuitive, it is conceivable that the XBB.1.5 RBD elicits antibodies that, for instance, due to increased affinity, are more effective in binding to heterologous molecules.

Chalkias et al., (https://www.medrxiv.org/content/10.1101/2023.08.22.23293434v2.full.pdf) recently published clinical trial results with the Moderna mRNA XBB.1.5 booster vaccine. That vaccine raised strong neutralizing antibody titers against XBB.1.5, but the ID50 values for neutralizing antibodies against BA.4/5 were even higher (6254 vs. 1639). Likewise, He et al., (2023) recently published that a BF.7 spike protein-based vaccine provided better protection against BA.5 pseudovirus than against the like BF.7 pseudovirus. In the same study, a similar observation was made for a BQ.1.1 antigen that protected better (higher IC50 value) against BA.5 than against the homologous BQ.1.1 pseudovirus.

Arguably, there are at least two possible explanations for this observation. Similar to live viruses, there are differences in viral infectivity between pseudoviruses with different spikes of variants of concern; an XBB.1.5 pseudovirus may be more or less effective in infecting target cells than a BA.4/5 pseudovirus. This will influence the calculated ID50 value. It is however also conceivable that XBB.1.5 elicits antibodies that indeed are more effective in blocking the entry of BA.4/5 pseudoviruses than even the homologous antigen would be. While we agree with the reviewer that this phenomenon requires attention, we note that the main purpose of the current manuscript was to demonstrate the functionality of the XBB-RBD vaccine against XBB.1.5 pseudoviruses. Understanding the binding profiles of the cross-reactive antibodies raised after XBB.1.5 vaccination is an ongoing project in our group.

In the manuscript edits, we have tried to better acknowledge this observation and edited the text in the results section for Figure 3B.

  1. In Figure 4, the neutralizing antibody titer of the BA4/5 RBD vaccine is missing. Please include that.

Response: Thank you for noticing this missing dataset in the figure.  The figure has been updated.

Round 2

Reviewer 3 Report

The authors have addressed all questions that I raised, and I recommend this manuscript to be accepted in the current form.

/